# Germination and Vigor of Soybean and Corn Seeds Treated with Mixed Mineral Fertilizers

**DOI:** 10.3390/plants12020338

**Published:** 2023-01-11

**Authors:** Carla Gomes Machado, Givanildo Zildo da Silva, Simério Carlos Silva Cruz, Rafael Cardoso Lourenço dos Anjos, Caíque Lopes Silva, Lucas Ferreira Lima de Matos, Alex Oliveira Smaniotto

**Affiliations:** Laboratory of Seeds, Federal University of Jataí (UFJ), Câmpus Jatobá—Cidade Universitária BR 364, Jataí 75801-615, Brazil

**Keywords:** seed treatment, micronutrients, physiological potential, seed quality, *Zea mays*, *Glycine max*

## Abstract

The use of mixed mineral fertilizers consisting of macro- and micronutrients, which are already routinely used for leaf fertilization, in the treatment of corn and soybean seeds would positively influence germination and vigor, consequently improving growth and seedling development. This study aimed to evaluate the physiological potential of corn and soybean seeds treated with mixed mineral fertilizers. The seed treatment efficiency test with mixed mineral fertilizers was conducted at the Federal University of Jataí—UFJ—using Vital^®^, Lança^®^, Ídolo^®^, Massivo^®^, and their combinations. The treatments consisted of two controls for soybean and three for corn. Analysis of variance was performed using the F-test (*p* ≤ 0.05), and when there was significance, the means were compared using the Scott–Knott test (*p* ≤ 0.05). The mixed mineral fertilizers Vital^®^ + Massivo^®^ (T6), Lança^®^ + Ídolo^®^ (T7), and Vital^®^ + Lança^®^ + Ídolo^®^ (T10) used in the seed treatment benefit the development of soybean (NS7667 IPRO) seeds. The treatments for corn (MG744 PWU) had little effect on germination and seedling development. The use of mixed mineral fertilizers in the seed treatments did not affect the physiological potential of the soybean and corn seeds, keeping the lots with germination values within commercialization standards.

## 1. Introduction

Adequate crop performance in the field is initially obtained through an ideal plant population, which depends on the correct use of several practices, among which the use of high-quality seeds treated with products that allow improved performance stands out [1].

Seed treatment is the application of fungicides, insecticides, inoculants, stimulants, biological products, or nutrients that preserve or improve seed performance, allowing crops to express their full genetic potential. It is a consolidated technique for cultivating crops, mainly soybean and corn, and avoiding possible losses resulting from the action of soil and shoot pests and the malformation of vegetative structures caused by a lack of nutrients in the early stages [2,3,4,5,6,7].

The technique of providing nutrients through seeds is mainly associated with the difficulty of performing this action across a total area due to the small amounts of micronutrients used during fertilization or via foliar once there is a need for making these elements available in the early stages of development when the foliar area is still small, resulting in poor application efficiency.

The application of macronutrients that perform fundamental functions at the beginning of a plant’s development has also been used for the purpose of providing all of the plant’s demands during the most critical stages of crop establishment to ensure that these macronutrients will not be lacking.

Micronutrients are essential and indispensable even though they are required in smaller quantities than macronutrients. Although the participation of micronutrients is small, a lack can result in significant production losses as they generally constitute the activators of enzymes as well as structural components, which may favor germination and seed vigor and, consequently, crop establishment [8,9,10,11,12,13].

The most supplied micronutrients in seeds are molybdenum (Mo), zinc (Zn), copper (Cu), and manganese (Mn).

Molybdenum is an essential component of two important plant enzymes, nitrogenase, and nitrate reductase. It is part of the enzyme nitrogenase, the most important enzyme in biological N_2_ fixation in all fixing organisms. Moreover, Mo functions as an electron donor in the reduction of nitrate to nitrite when catalyzed by nitrate reductase [14]. Consequently, an inadequate supply of Mo will affect not only the specific functions of this element but also some of those related to nitrogen (N) [15].

Zinc acts as an activator or component of enzymes and participates in C4 plant photosynthesis through the enzyme pyruvic carboxylase, which is necessary for the production of tryptophan, amino acids, and plant hormones. Furthermore, it is also related to N metabolism [16]. Since its deficiency affects the synthesis and conservation of auxins, plant hormones involved in growth, the lack of this element slows seedling development, reducing the number of plants produced on the tillage [17].

Copper is a component of several enzymes, found predominantly in chloroplasts and part of plastocyanin. Mn plays a fundamental role in cell elongation; participates as a catalyst in enzymatic activities, chlorophyll synthesis, and photosynthesis; and its deficiency can inhibit the synthesis of lipids or secondary metabolites. The functions of enzyme activation, biosynthesis, energy transfer, and hormonal regulation attributed to Mn are fundamental to the formation, development, and maturation of seeds. This way, Mn, due to its nature, may be directly or indirectly involved with the physiological quality of seeds [18,19,20].

Among macronutrients, N stands out as an essential nutrient and is mainly required by most agricultural cultures. It is essential for plant growth and development and is also a constituent of amino acids, which are the main components of proteins, nucleic acids, nucleotides, and coenzymes [18,21].

Another macronutrient that deserves attention, mainly during the early stages of establishing crop fields, is phosphorous (P). This element is involved with almost all plant processes that involve energy transfer, boosting many chemical reactions. To ensure proper early seedling development, phosphorous is commonly provided by coating the seeds to achieve an optimal yield [22]. The supply of P to seeds was also found to be efficient in increasing germination, the emergence speed index, vigor, shoot dry matter, root freshness, and dry weight.

According to [23], balanced mineral nutrition, especially regarding N, Cu, and Mn, can attenuate the severity of diseases, as these nutrients participate in the defense mechanisms of plants.

Thus, this study hypothesized that the use of mixed mineral fertilizers consisting of macro- and micronutrients that are already routinely used for leaf fertilization in the treatment of corn and soybean seeds would positively influence germination and vigor, consequently improving growth and seedling development.

In this context, this study aimed to evaluate the physiological potential of corn and soybean seeds treated with mixed mineral fertilizers.

## 2. Results

### 2.1. Treated Soybean Seeds

The germination of the treated soybean seeds (Table 1) showed that most treatments did not differ from the control; that is, they did not affect germination. Germination was lower than the control only when Ídolo was present in the treatment composition; that is, T4: Ídolo; T6: V + L + I; T9: V + I; and T10: L + I, except for the treatments with Massivo in their composition (T8: V + M + I and T11: I + M), the latter two being superior to the four mentioned above, not differing from the others, including the control.

A percentage of strong normal seedlings (Ns) above 72% was found in the control treatments with intact seeds, T1: Vital (V); T2: Lança (L); T3: Massivo (M); T5: V + L; and T8: V + M + I. No differences were found between the treatments and controls for the variable abnormal seedlings and the dead seeds after the germination test, as well as in the evaluation of the seedlings in the sand in relation to emergence, first count, and speed index; that is, the seed treatment did not interfere with these quality aspects.

The shoot lengths of the soybean seedlings in most treatments did not differ from the controls (Table 2), except for the treatment of T5: V + L; T9: V + I; and T11: I + M. However, their dry matter presented higher stratification, with most treatments being superior to the intact seed control. However, no treatment harmed the shoot dry matter to the point of being inferior to the seeds normally used. The main root lengths of the soybean seedlings showed a behavior similar to that of the shoot; that is, most of the treatments did not differ from the controls, except the treatment of T5: V + L, L; T9: V + I; and T11: I + M. In contrast, the root dry matter was superior for the seeds treated with the combinations of T6: V + L + I; T7: V + M; and T10: L + I, while the other treatments did not differ from the controls.

The use of Massivo and Lança alone in the treatment of the soybean seeds, as well as the associations, T5: V + L; T6: V + L + I; T7: V + M; T8: V + M + I; T10: L + I; and T11: I + M, provided a higher emergence and seedling emergence speed index in the field, even being superior to the controls (Table 3).

The other variables that evaluated the emergence speed showed a difference in the first emergence count, with behavior similar to emergence, except for the product Lança and T8: V + M + I. A difference was also observed in terms of the emergence speed index, with the treatments of T2: Lança; T3: Massivo; T5: V + L; T6: V + L + I; T7: V + M; T8: V + M + I; T10: L + I; and T11: I + M, which were superior to the controls and T1: Vital; T4: Ídolo; and T9: Vital + Ídolo.

Figure 1 shows the relative frequency of seedling emergence in the field. Most treatments had unimodal behavior; that is, the seedlings emerged until they reached a maximum value and then declined, except those treated with T1: Vital; T4: Ídolo; T5: V + L; and T7: V + M, for which the maximum is reached, then declined, and increased again, but with lower peaks than the initial one, showing a polymodal behavior; that is, more than one seedling emergence peak.

In general, all of the frequency polygons exhibited a homogeneous distribution of emergence, even in the treatments in which more than one peak was observed, as emergence occurred between 3 and 7 days after sowing, a result that confers speed and uniformity in seedling formation. This homogeneity is corroborated by similar values for seedling emergence at an initial time of 4 days, an average time of 5 days, and a final time between 7 and 8 days (Table 3).

Two components with total variances of 34.3 and 26.3% were required to explain the data variability in the multivariate principal component analysis of the soybean seed treatments (Figure 2). The sum of these values reached 60.5% of the accumulated variance.

According to the PCA biplot, three groups of soybean seeds were formed in consideration of the different treatments. Group 1 was predominated by the treatments that provided higher seedling emergence in the sand substrate and in the field and better performance in growth and dry matter accumulation, which are T6: V + L + I; T7: V + M; and T10: L + I.

The control treatments, the intact seeds, and the seeds only treated with water, in addition to T1: Vital^®^; T2: Lança^®^; T3: Massivo^®^; T5: V + L; T8: V + M + I; T9: V + I; and T11: I + M were classified as intermediate, i.e., the treatments close to the origin of the axes [24]. Furthermore, the values of germination, emergence, and speeds of these processes stood out but were similar to the controls. It denotes that the aforementioned treatments at least maintain the quality of the soybean seed lot.

Finally, the treatments of T4: Ídolo^®^ and T9: V + I provided a lower performance in relation to the soybean seeds, with decreased values of germination, emergence, and the development of the resulting seedlings. Thus, it shows that the composition incorporating manganese can be harmful, but it is attenuated when using a treatment with Massivo^®^.

### 2.2. Treated Corn Seeds

The treated corn seeds (T1 to T11) had percentages of germination similar to the controls; that is, they did not affect germination, except for the seeds treated with Lança (T2), which presented lower germination than the control (Table 4).

The highest percentages of strong normal seedlings were obtained from the seeds treated with T1: Vital; T3: Massivo; T4: Ídolo; T7: V + M; T8: V + M + I; T10: L + I; and T11: I + M. These treatments were even superior to the controls, which did not differ from the treatments of T5: V + L and T9: V + I.

The highest percentages of abnormal seedlings and dead seeds were verified in the seeds submitted to the treatment of T2: Lança, with the first variable being the washed control seeds and water added to the treatment, as well as those from T2: Lança, which exhibited a similar behavior for the percentage of abnormal seedlings.

The seedling emergence in sand presented no differences between the treatments. However, the emergence speed index showed differences between the treatments. The treatments of T2: Lança; T3: Massivo; T4: Ídolo; T7: V + M; and T9: V + I were similar to the control intact seeds, where the seeds were washed and added to water regarding the first count; these treatments were superior. However, the treatments grouped as inferior were similar to the control in which the seeds were only washed.

Higher shoot lengths of the corn seedlings were obtained from the seeds treated with T1: Vital; T2: Lança; T3: Massivo; T4: Ídolo; T6: V + L + I; and T7: V + M, which did not differ from the controls (Table 5). The seedlings measured in the treatments of T5: V + L; T8: V + M + I; and T9: V + I were classified as intermediate. The seeds treated with T10: L + I and T11: I + M presented a delay in the shoot development of the seedlings. These last treatments, plus T8: V + M + I and T9: V + I, were also harmful to the shoot dry matter accumulation of seedlings. In contrast, the seeds treated with T7: V + M provided higher shoot dry matter accumulation, while the other treatments were intermediate.

Regarding the main root length, T1: Vital; T3: Massivo; T7: V + M; and T9: V + I were the only treatments that did not differ from one of the controls. The other treatments had the main root development delayed, but the treatments T2: Lança; T5: V + L; T10: L + I; and T11: I + M affected the dry matter accumulation in the root system. Importantly, only the length of the main root or the largest fibrous root is evaluated, while the dry matter consists of the entire root system. The root dry matter also showed a higher accumulation when the seeds were treated with T3: Massivo. This behavior was similar in the root-to-shoot ratio, with the addition of the treatments T10: L + I and T11: I + M.

The seeds with higher vigor after the cold test, which did not differ from the controls, were obtained from treatments T1: Vital; T3: Massivo; T4: Ídolo; T5: V + L, T9: V + I; and T11: I + M.

While corn seedling emergence in the field showed no difference between the treatments across all of the variables (Table 6), this homogeneity of the results shows that the treatments maintained the high initial quality of the seed lot.

The relative frequency of corn seedling emergence in the field shows that all of the treatments provided a unimodal behavior; that is, seedlings emerged until they reached a maximum value and then declined (Figure 3). Therefore, all of the frequency polygons presented a homogeneous distribution of emergence, as the emergence was between 3 and 6 days after sowing, a result that confers speed and uniformity in seedling formation. This homogeneity is corroborated by similar values in seedling emergence at an initial time of 4 days, an average time of approximately 5 days, and a final time between 5 and 6 days (Table 6). The homogeneity of frequency of the seedling emergence obtained from treated seeds allows for an adequate initial stand, thus demonstrating the potential for the TS of these products and their combinations.

The set of 12 variables in the evaluation of the quality of corn seeds treated or not with mixed products, isolated or associated, allowed a principal component analysis to synthesize the maximum of the original information in two latent variables called principal components, enabling its location in two-dimensional plots (ordering of accessions of principal components), as shown in Figure 4.

The division into three groups was also observed in corn seeds from different treatments. However, the treatment with most products and their associations did not result in a significant difference in germination and evaluated seedlings, as they are in Group 1, together with the three controls used in the study. Group 2 was formed by the treatments T210: L + I and T11: I + M, as they presented higher values in the root-to-shoot ratio. On the other hand, T2: Lança^®^ was in Group 3 for providing higher values of abnormal seedlings and dead seeds, but only 3%, maintaining a 94% germination rate.

The treatments used in the present study did not negatively affect corn seeds. At this point, we can highlight some treatments that are more distant from the origin: the stand-out results are the higher values of vigor by the cold test and seedling formation (T3: Massivo^®^, T4: Ídolo^®^, and T7:V + M), as well as germination and the root-to-shoot dry matter accumulation ratio (T10: L + I and T11: I + M).

## 3. Discussion

Based on the germination results relating to soybeans (NS7667 IPRO), the composition with manganese can impair germination, but the association with Massivo, a product containing zinc, can attenuate this deleterious effect. Importantly, all of the treatments provided germination rates higher than 80%, which can be classified as seeds according to the standard for the production and commercialization of seeds in Brazil [25]. Additionally, the best treatments provided germination close to 90%, and most of them were numerically higher than the controls.

There was no difference between treatments in the variable’s abnormal seedlings and the dead seeds in germination, as well as seedlings in the sand by emergence, and the first count and speed index are extremely relevant since the large-scale use of products requires previous studies to verify their effectiveness and possible harmful effects on the physiological seed quality [26].

The higher shoot and root length can influence the later crop development stages, as more developed roots allow better fixation in the soil and the absorption of available nutrients, and more developed leaves have better photosynthetic efficiency [18].

In the climatic description, Figure 5 shows that these results are closely related to the treatments, as the seedling emergence environment in the field had a temperature between 20 and 30 °C, in addition to precipitation and supplementary irrigation according to what is required for soybean cultivation [27]. The homogeneity of the seed treatments allows a uniform emergence and, consequently, a uniform and adequate initial seedling stand, as the seeds had adequate germination.

The principal component analysis of the soybean seed treatments was adequate, as the values of the principal components were similar to those found in studies evaluating soybean seed lots [28] and safflower vigor tests [29,30]. According to [31], a relatively small number of components were extracted (PC1 and PC2), with the ability to explain the highest variability in the original data.

In the evaluation of the treated corn (MG744 PWU) seeds, it was verified that the T10 treatment, represented by the association between the products Lança (consisting of N: 65.10 g L^−1^; Cu: 304.50 g L^−1^; Mo: 84.00 g L^−1^; and Zn: 619.50 g L^−1^) and Ídolo (N: 63.55 g kg^−1^ and Mn: 647.80 mg kg^−1^), compromised corn development. According to [32], adequate N, Mn, and Zn contents present in corn seed are 13.9 g kg^−1^, 4.80 mg kg^−1^, and 17.3 mg kg^−1^, respectively. The N, Mn, and Zn contents present in the corn seed used in the present study were 18.9 g kg^−1^, 4.80 mg kg^−1^, and 22.3 mg kg^−1^ (Table 7), respectively, that is, the N, Mn, and Zn concentrations were above the range considered adequate for corn seeds. Therefore, the use of these products led to excess, compromising the development.

The same behavior was observed in treatment T11, represented by the use of the products Ídolo (consisting of N: 63.55 g kg^−1^ and Mn: 647.80 mg kg^−1^) and Massivo (N: 30.24 g kg^−1^ and Zn: 1466.64 mg kg^−1^). The seeds already had levels above what is considered ideal, and the use of these products led to excess, resulting in a negative effect on shoot length, shoot dry matter, and root length.

Phytotoxicity in seeds commonly manifests in the root length [33]. Most of the products that interfered with the development had manganese in their composition. Manganese is involved in plant enzymatic systems either as a constituent (cofactor) or as an enzyme activator. Therefore, the functions of enzymatic activation, biosynthesis, energy transfer, and hormonal regulation are essential for seed formation, development, and maturation [20]. In this sense, manganese, by its nature, may be directly or indirectly involved in the physiological quality of the produced seeds.

This homogeneity of the results relating to corn seedling emergence shows that the treatments maintained the high initial quality of the seed lot. Thus, these treatments are suitable for seed treatment both in industry and on the farm, carried out by the producers. According to [26], a given seed treatment is successful and can be commercially adopted as long as it does not cause a harmful effect on the physiological seed quality.

Water was supplied by irrigation in the emergence test because the precipitation during the experimental period proved to be inadequate to meet the requirements of the corn seedlings. The temperature ranged from 5 to 30 °C (Figure 5), being out of the ideal range for seed germination [27].

In the principal components analysis of the treated corn seeds, it was verified that the total variance accumulated by each component was 33.7 and 31.1%, totaling 64.7%. These values are similar to those found in studies on the physiological and sanitary quality of *Urochloa brizantha* cv. BRS Piatã [34,35].

The multivariate data analysis methods allow a global study of these variables, evidencing the connections, similarities, or differences between them with the least possible loss of information [26,29,30,31,34].

The data collection and analysis carried out during the experiments were mostly conducted under lab-controlled conditions to provide reliability; however, the scope of these results is limited to the growing of soybeans and the hybrid corn used in these experiments.

## 4. Methodology

This study refers to the trial of seed treatment efficiency with mixed mineral fertilizers conducted by the Federal University of Jataí (UFJ) with Vital^®^, Lança^®^, Ídolo^®^, and Massivo^®^.

The seeds were treated with the mineral fertilizer Vital^®^, which is composed of phosphorus (P_2_O_5_ 10.2 g L^−1^) and molybdenum (Mo 10.2 g L^−1^). The mineral fertilizer Lança^®^ consists of nitrogen (N 65.1 g L^−1^), copper (Cu 304 g L^−1^), molybdenum (Mo 84.0 g L^−1^), and zinc (Zn 619 g L^−1^). The commercial product Massivo^®^ is composed of nitrogen (30.2 g L^−1^) and zinc (1466 g L^−1^). The mineral fertilizer Ídolo^®^ consists of nitrogen (63.5 g L^−1^) and manganese (647 g L^−1^).

The mixed mineral fertilizers Lança, Massivo, and Ídolo are products based on cupric oxide, zinc oxide, and manganese carbonate, respectively.

The experiments were carried out in the Laboratory of Seeds and the experimental area of the Fazenda Escola at the Federal University of Jataí. Soybean NS7667 IPRO and corn MG744 PWU seeds were used. Table 7 shows the chemical analysis of the macro- and micronutrients in the soybean and corn seeds.

The initial characterization of soybean NS7667 IPRO and corn MG744 PWU seeds was carried out using the variables of the thousand-seed weight (TSW), water content (WC), germination (G), abnormal seedlings (AS), and dead seeds (DS), according to the Rules for Seed Testing [27], used in the experiments.

Soybean seeds had a TSW of 191.78 g, WC of 10.2%, G of 89%, 5% of AS, and 6% of DS, while corn seeds had a TSW of 428.24 g, WC of 10.2%, G of 99%, 0% of AS, and 1% of DS. Importantly, both soybean and corn seeds were above the standard of production and commercialization of seeds in Brazil [25].

Subsequently, the seeds were subjected to treatments with mixed mineral fertilizers (Table 8), with a solution volume corresponding to the dose of 500 mL for 60,000 corn seeds or 100 kg of soybean seeds.

The amount of fertilizer is commonly related to the mass of seeds in studies involving seed treatment. Corn seeds are usually marketed considering their quantity; therefore, it is convenient to recommend this type of treatment per seed unit, as a bag with sixty thousand corn seeds can vary greatly according to their mass, mainly depending on the sieve [26].

Control treatments were used for each species. Intact white seeds and only the solution with water were used for soybean, totaling two controls. Corn seeds had an industrial treatment, and, therefore, the controls consisted of intact seeds (industrial treatment), washed seeds, and washed seeds plus a solution with water, totaling three controls. The washing process consisted of adding 200 mL of distilled water to containers with seeds, followed by a slight manual shaking for two minutes, transference to sieves to be washed in running distilled water with 200 mL, and placing on a paper towel for superficial drying for 30 min, on average.

Eleven treatments were used, in addition to those mentioned, for which samples of 1 kg of soybean seeds and 960 corn seeds were chemically treated (Table 8). The products were added to the seeds using graduated syringes (mm). The seeds were shaken for two minutes in plastic bags for homogenization after the addition of each product [36].

### 4.1. Evaluation of the Physiological Quality of Seeds

The following determinations and evaluations were performed after the treatments:
Germination test: The test was conducted in a completely randomized design in the Laboratory of Seeds, with 8 subsamples of 50 seeds from each treatment, totaling 400 seeds. The seeds were sown in 136 g of autoclaved medium-textured dry sand arranged in a Germitest^®^ paper towel roll previously moistened with distilled water in an amount equivalent to 3.0 times the weight of the dry substrate and maintained in a germination chamber regulated at 25 °C [27,37]. The final evaluation was performed at 8 and 7 days after sowing for soybean and corn, respectively, by counting normal and abnormal seedlings and dead seeds. The interpretations were performed according to the criteria established in the Rules for Seed Testing [27]. Moreover, normal seedlings were classified as strong normal according to [38].Seedling length: This was carried out in a completely randomized design with eight replications. Twenty seeds per lot were sown on two parallel lines drawn in the upper third of the paper between 54 g of autoclaved medium-textured dry sand [37]. The paper had been previously moistened with an equivalent amount of water to 3.0 times its weight. The paper towel rolls were placed in plastic bags to reduce dehydration and maintained at 25 °C, being interrupted 6 days after sowing. After this period, the seedlings were measured in terms of root and shoot length using a common ruler graduated in mm. Subsequently, they were placed in kraft paper bags and taken to a forced ventilation oven at 65 °C to determine the shoot and root dry matter, as well as the root-to-shoot ratio [38].Seedling emergence in sand: This was carried out in six randomized blocks with 25 seeds each, which were sown in moistened sand at 60% of the substrate water retention capacity, calculated according to the Rules of Seed Testing [27], inside transparent acrylic boxes (11.0 × 11.0 × 3.5 cm) maintained on a laboratory bench at 26 ± 3 °C. The percentage of emerged seedlings was counted after the stabilization of counts.Seedling emergence in the field: This was conducted in five randomized blocks. Sowing was carried out in 50 cm beds. The seeds were arranged in a double row at a depth of 5 cm, with 25 seeds in each row, totaling 250 seeds per treatment. Daily counts were performed by counting the number of emerged seedlings (total emergence of cotyledons for soybean and coleoptile with emission of primary leaves for corn), until reaching stabilization. Daily watering with micro-sprinklers was carried out during the experimental period to maintain moisture. Precipitation and temperatures were also monitored (Figure 5).Emergence speed index (ESI): This was conducted simultaneously with the emergence tests performed in the sand and the field from the daily count data of the seedlings that emerged in the seedling emergence tests and determined using the formula from [39].First count of seedling emergence: This was conducted together with the emergence tests in the sand and the field with an evaluation on the fifth and fourth day after sowing of soybean and corn, respectively, in which emergence higher than 50% + 1 of the final count was observed.Initial, final, and average time and relative frequency of seedling emergence: This was determined based on the daily count data of seedlings that emerged from the seedling emergence tests. The initial time corresponds to the first day of emergence after sowing, and the final time consists of the last day of emergence after sowing for each lot. The formulas described by the authors of [40] were used for average time and relative frequency of emergence.

### 4.2. Statistical Analysis

The data were subjected to normality and homogeneity tests and transformed into (x + 1)1/2, if necessary, but the original data were presented. Analysis of variance was performed using the F-test (*p* ≤ 0.05), and the means were compared using the Scott–Knott test (*p* ≤ 0.05) when significance was observed. The AgroStat^®^ software [41] was used for analysis.

## 5. Conclusions

The the mixed mineral fertilizers Vital^®^ + Massivo^®^ (T6), Lança^®^ + Ídolo^®^ (T7), and Vital^®^ + Lança^®^ + Ídolo^®^ (T10) used in the seed treatment benefit the development of soybean (NS7667 IPRO) seeds. The treatments for corn (MG744 PWU) had little effect on germination and seedling development.

The use of mixed mineral fertilizers in seed treatment did not affect the physiological potential of soybean and corn seeds, keeping the lots with germination values within the commercialization standard.

## Figures and Tables

**Figure 1 plants-12-00338-f001:**
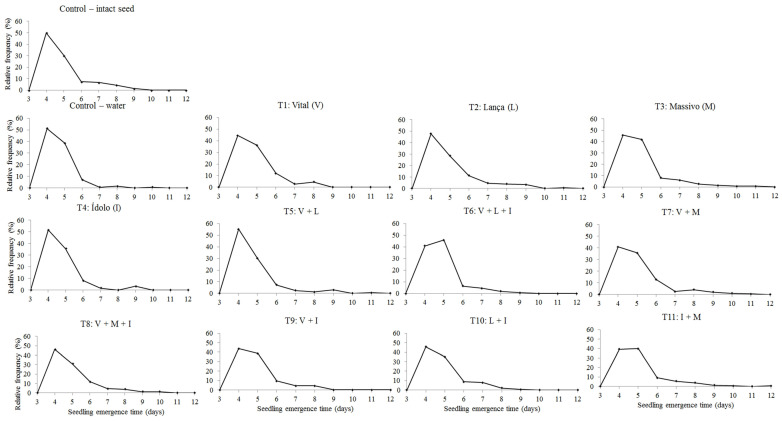
Distribution of relative frequency of soybean NS7667 IPRO seedling emergence in the field from different seed treatments. Jataí, GO, Brazil, 2022.

**Figure 2 plants-12-00338-f002:**
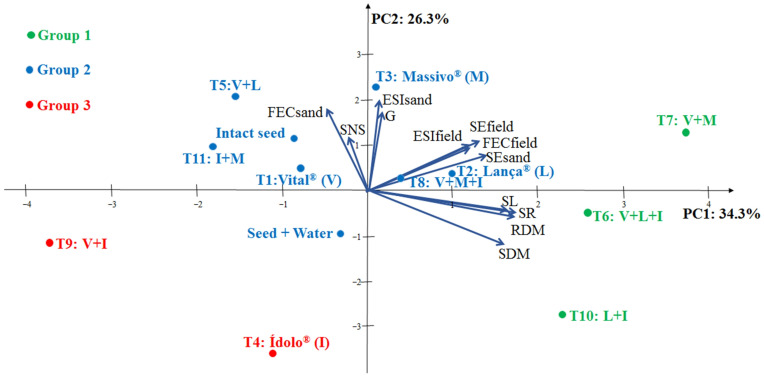
Biplot with the circle of eigenvectors obtained by the analysis of two principal components (PC1 and PC2) established according to germination (G), strong normal seedling (SNS), seedling emergence in sand (SEsand), emergence speed index in sand (ESIsand), first emergence count in sand (FECsand), seedling emergence in the field (SEfield), emergence speed index in the field (ESIfield), first emergence count in the field (FECfield), root length (RL), shoot length (SL), root dry matter (RDM), and shoot dry matter (SDM) of soybean NS7667 IPRO seedlings from different seed treatments. Jataí, GO, Brazil, 2022.

**Figure 3 plants-12-00338-f003:**
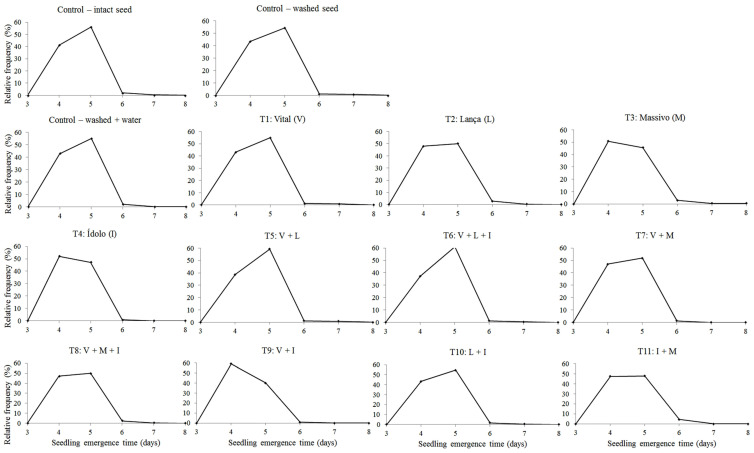
Distribution of relative frequency of corn MG744 PWU seedling emergence in the field from different seed treatments. Jataí, GO, Brazil, 2022.

**Figure 4 plants-12-00338-f004:**
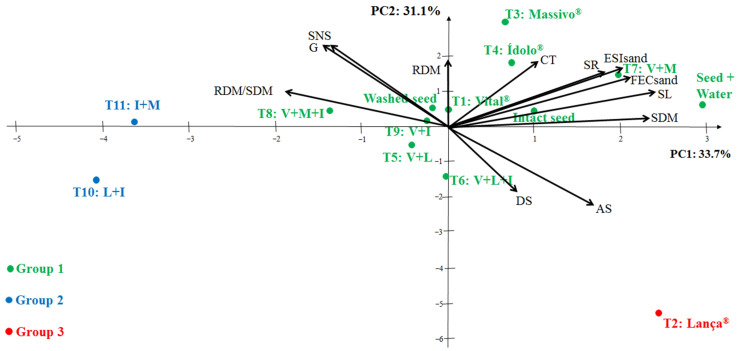
Biplot with the circle of eigenvectors obtained by the analysis of two principal components (PC1 and PC2) established according to germination (G), strong normal seedling (SNS), abnormal seedlings (AS), dead seeds (DS), first emergence count (FECsand), seedling emergence speed index in sand (ESIsand), shoot length (SL), shoot dry matter (SDM), root length (SR), root dry matter (RDM), root-to-shoot dry matter ratio (RDM/SDM), and cold test (CT) of corn MG744 PWU from different seed treatments. Jataí, GO, Brazil, 2022.

**Figure 5 plants-12-00338-f005:**
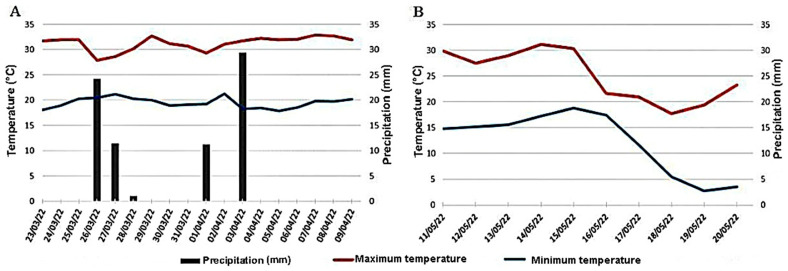
Maximum and minimum temperatures and precipitations during the period when the seedling emergence tests of soybean (**A**) and corn (**B**) were conducted in the field. Jataí, GO, Brazil, 2022. Source: INMET (2022).

**Table 1 plants-12-00338-t001:** Germination (G), strong normal seedlings (Ns), abnormal seedlings (AN), dead seeds (DS), seedling emergence in sand (Es), first count (FCEs), and seedling emergence speed index in sand (ESIs) of soybean NS7667 IPRO from different seed treatments. Jataí, GO, Brazil, 2022.

Treatment	G	Ns	AN	DS	Es	FCEs	ESIs
	%						
Control—intact seed	88.0 a ^1^	74.0 a	10.0	2.00	85.0	44.0	3.69
Control—water	88.0 a	69.0 b	9.00	3.00	86.0	39.0	3.26
T1: Vital^®^ (V)	91.0 a	83.0 a	7.00	3.00	89.0	44.0	3.47
T2: Lança^®^ (L)	89.0 a	73.0 a	7.00	4.00	89.0	38.0	3.51
T3: Massivo^®^ (M)	89.0 a	78.0 a	10.0	2.00	90.0	43.0	3.44
T4: Ídolo^®^ (I)	80.0 b	67.0 b	16.0	4.00	83.0	35.0	2.98
T5: V + L	88.0 a	72.0 a	10.0	2.00	86.0	40.0	3.54
T6: V + L + I	83.0 b	67.0 b	14.0	4.00	88.0	42.0	3.46
T7: V + M	89.0 a	68.0 b	8.00	3.00	89.0	41.0	3.43
T8: V + M + I	90.0 a	75.0 a	8.00	2.00	85.0	41.0	3.28
T9: V + I	85.0 b	67.0 b	11.0	4.00	83.0	40.0	3.26
T10: L + I	85.0 b	66.0 b	12.0	4.00	87.0	31.0	3.17
T11: I + M	87.0 a	66.0 b	11.0	2.00	81.0	44.0	3.41
Mean square—Treatment	70.2 *	216 **	51.5 ns	6.80 ns	44.8 ns	84.4 ns	0.100 ns
Mean square—Blocks	-	-	-	-	41.5 ns	20.6 ns	0.00 ns
Mean square—Residual	35.8	76.8	29.1	6.00	42.3	122	0.100
Coefficient of variation (%)	6.80	12.3	53.3	85.0	7.05	27.5	10.0

^1^ Means followed by the same letter in the column do not differ from each other when using the Scott–Knott clustering test at a 5% probability. **, *, and ns significant at 1 and 5% probability and not significant in relation to the F-test, respectively.

**Table 2 plants-12-00338-t002:** Average shoot length (SL), shoot dry matter (SDM), root length (RL), root dry matter (RDM), and root-to-shoot ratio (R/S) of soybean NS7667 IPRO seedlings from different seed treatments. Jataí, GO, Brazil, 2022.

Treatment	SL (cm)	SDM (mg)	RL (cm)	RDM (mg)	R/S
Control—intact seed	6.40 a ^1^	18.5 b	8.30 a	9.20 b	0.496
Control—water	6.80 a	19.5 a	8.50 a	9.50 b	0.486
T1: Vital^®^ (V)	7.10 a	20.5 a	8.80 a	9.70 b	0.471
T2: Lança^®^ (L)	7.30 a	20.7 a	8.20 a	10.20 b	0.487
T3: Massivo^®^ (M)	6.10 a	16.2 b	8.30 a	8.30 b	0.512
T4: Ídolo^®^ (I)	6.60 a	20.3 a	8.40 a	7.10 b	0.349
T5: V + L	4.90 b	16.6 b	5.50 b	5.70 b	0.332
T6: V + L + I	7.30 a	21.2 a	9.30 a	18.9 a	0.966
T7: V + M	7.40 a	21.4 a	10.80 a	12.6 a	0.583
T8: V + M + I	6.50 a	21.0 a	9.20 a	9.30 b	0.459
T9: V + I	4.70 b	15.6 b	6.50 b	6.90 b	0.415
T10: L + I	6.90 a	24.0 a	9.50 a	14.9 a	0.661
T11: I + M	5.50 b	16.4 b	7.00 b	5.70 b	0.341
Mean square—Treatment	6.58 **	5.12^−5^ **	15.5 **	1.12^−4^ *	0.200 ns
Mean square—Residual	1.40	1.66^−5^	3.60	4.92^−5^	0.100
Coefficient of variation (%)	18.5	21.0	22.7	71.3	76.4

^1^ Means followed by the same letter in the column do not differ from each other when using the Scott–Knott clustering test at a 5% probability **, *, and ns significant at 1 and 5% probability and not significant when using the F-test, respectively.

**Table 3 plants-12-00338-t003:** Seedling emergence in the field (Ef), first count (FCEf), and seedling emergence speed index in the field (ESIf) at initial, average, and final emergence times of soybean NS7667 IPRO seedlings in the field from different seed treatments. Jataí, GO, Brazil, 2022.

Treatment	Ef	FCEf	ESIf	Tif	Taf	Tff
	%			Days		
Control—intact seed	54.0 b ^1^	44.0 b	2.91 b	4.00	5.00	8.00
Control—water	51.0 b	46.0 b	2.81 b	4.00	5.00	7.00
T1: Vital^®^ (V)	43.0 b	35.0 b	2.30 b	4.00	5.00	7.00
T2: Lança^®^ (L)	60.0 a	46.0 b	3.18 a	4.00	5.00	8.00
T3: Massivo^®^ (M)	64.0 a	53.0 a	3.40 a	4.00	5.00	8.00
T4: Ídolo^®^ (I)	50.0 b	43.0 b	2.72 b	4.00	5.00	7.00
T5: V + L	65.0 a	56.0 a	3.58 a	4.00	5.00	8.00
T6: V + L + I	63.0 a	54.0 a	3.35 a	4.00	5.00	7.00
T7: V + M	77.0 a	59.0 a	4.02 a	4.00	5.00	8.00
T8: V + M + I	61.0 a	47.0 b	3.21 a	4.00	5.00	8.00
T9: V + I	47.0 b	38.0 b	2.49 b	4.00	5.00	7.00
T10: L + I	60.0 a	49.0 a	3.22 a	4.00	5.00	7.00
T11: I + M	65.0 a	52.0 a	3.39 a	4.00	5.00	8.00
Mean square—Treatment	860 **	483 **	2.10 **	-	0.100 ns	3.00 ns
Mean square—Blocks	2536 **	2978 *	8.10 **	-	0.300 ns	1.30 ns
Mean square—Residual	193	148	0.500	-	0.100	2.00
Coefficient of variation (%)	23.7	25.4	23.7	-	7.5	19.2

^1^ Means followed by the same letter in the column do not differ from each other when using the Scott–Knott clustering test at a 5% probability **, *, and ns significant at 1 and 5% probability and not significant when using the F-test, respectively.

**Table 4 plants-12-00338-t004:** Germination (G), strong normal seedlings (Ns), abnormal seedlings (AN), dead seeds (DS), seedling emergence in sand (Es), first count (FCEs), and seedling emergence speed index in sand (ESIs) of corn MG744 PWU from different seed treatments. Jataí, GO, Brazil, 2022.

Treatment	G	Ns	AN	DS	Es	FCEs	ESIs
	%						
Control—intact seed	99.0 a ^1^	87.0 b	1.00 b	0.00 b	100	91.0 a	7.03 b
Control—washed	100 a	89.0 b	0.00 b	0.00 b	97.0	87.0 b	6.78 c
Control—washed + water	98.0 a	87.0 b	2.00 a	0.00 b	100	95.0 a	7.35 a
T1: Vital^®^ (V)	99.0 a	95.0 a	0.00 b	1.00 b	100	85.0 b	6.96 c
T2: Lança^®^ (L)	94.0 b	74.0 c	3.00 a	3.00 a	100	89.0 a	6.91 c
T3: Massivo^®^ (M)	99.0 a	94.0 a	0.00 b	1.00 b	100	93.0 a	7.28 a
T4: Ídolo^®^ (I)	100 a	95.0 a	0.00 b	0.00 b	99.0	96.0 a	7.09 b
T5: V + L	99.0 a	87.0 b	1.00 b	0.00 b	99.0	87.0 b	6.93 c
T6: V + L + I	99.0 a	81.0 c	1.00 b	0.00 b	99.0	85.0 b	6.70 d
T7: V + M	99.0 a	94.0 a	0.00 b	1.00 b	100	95.0 a	7.28 a
T8: V + M + I	100.0 a	95.0 a	0.00 b	0.00 b	99.0	88.0 b	6.90 c
T9: V + I	99.0 a	88.0 b	1.00 b	0.00 b	99.0	91.0 a	7.03 b
T10: L + I	99.0 a	93.0 a	0.00 b	1.00 b	98.0	82.0 b	6.63 d
T11: I + M	100 a	97.0 a	0.00 b	0.00 b	99.0	86.0 b	6.86 c
Mean square—Treatment	15.3 **	322 **	7.00 **	2.90 **	4.80 ns	154 **	0.300 **
Mean square—Blocks	-	-	-	-	2.70 ns	25.9 ns	0.200 *
Mean square—Residual	3.36	56.5	2.10	1.26	4.10	34.2	0.100
Coefficient of variation (%)	1.80	8.30	240	175	2.00	6.60	3.60

^1^ Means followed by the same letter in the column do not differ from each other when using the Scott–Knott clustering test at a 5% probability **, * and ns significant at 1 and 5% probability and not significant when using the F-test, respectively.

**Table 5 plants-12-00338-t005:** Average shoot length (SL), shoot dry matter (SDM), root length (RL), root dry matter (RDM), and root-to-shoot ratio (R/S) of corn seedlings and cold test (CT) of corn MG744 PWU seeds from different seed treatments. Jataí, GO, Brazil, 2022.

Treatment	SL (cm)	SDM (mg)	RL (cm)	RDM (mg)	R/S	CT (%)
Control—intact seed	8.50 a ^1^	25.2 b	13.8 a	36.7 b	1.04 b	99.0 a
Control—washed	8.60 a	33.2 b	12.0 a	40.6 b	1.21 b	99.0 a
Control—washed + water	9.00 a	36.3 b	13.8 a	40.1 b	1.11 b	99.0 a
T1: Vital^®^ (V)	8.90 a	34.7 b	12.7 a	51.0 b	1.45 b	96.0 a
T2: Lança^®^ (L)	7.80 a	34.8 b	9.10 c	26.0 b	0.75 b	93.0 b
T3: Massivo^®^ (M)	9.30 a	33.3 b	12.8 a	105 a	3.13 a	99.0 a
T4: Ídolo^®^ (I)	9.70 a	31.6 b	11.3 b	43.3 b	1.37 b	98.0 a
T5: V + L	6.70 b	33.4 b	7.90 d	44.5 b	1.33 b	99.0 a
T6: V + L + I	8.70 a	37.6 b	10.8 b	48.7 b	1.30 b	93.0 b
T7: V + M	9.70 a	44.9 a	12.2 a	59.4 b	1.34 b	95.0 b
T8: V + M + I	6.70 b	27.1 c	10.7 b	39.0 b	1.44 b	95.0 b
T9: V + I	5.50 b	24.4 c	12.5 a	40.7 b	1.69 b	98.0 a
T10: L + I	4.40 c	18.5 d	8.70 c	50.5 b	2.98 a	90.0 b
T11: I + M	3.60 c	16.6 d	7.30 d	40.8 b	2.51 a	97.0 a
Mean square—Treatment	15.4 **	2.29^−4^ **	18.1 **	1.35^−3^ **	2.10 **	46.5 **
Mean square—Residual	0.70	1.16^−5^	1.00	2.61^−4^	0.100	7.74
Coefficient of variation (%)	11.5	10.8	9.00	26.4	24.6	2.88

^1^ Means followed by the same letter in the column do not differ from each other when using the Scott–Knott clustering test at a 5% probability ** Significant at 1% probability when using the F-test.

**Table 6 plants-12-00338-t006:** Seedling emergence in the field (Ef), first count (FCEf) and seedling emergence speed index in the field (ESIf), abnormal seedlings (ANf), and initial, average, and final times of seedling emergence in the field of corn MG744 PWU from different seed treatments. Jataí, GO, Brazil, 2022.

Treatment	Ef	FCEf	ESIf	Tif	Taf	Tff
	%			Days		
Control—intact seed	100 ^1^	97.0	5.47	4.00	4.60	6.00
Control—washed	98.0	96.0	5.43	4.00	4.60	5.00
Control—washed + water	100	98.0	5.50	4.00	4.60	5.00
T1: Vital^®^ (V)	98.0	96.0	5.38	4.00	4.60	6.00
T2: Lança^®^ (L)	99.0	96.0	5.50	4.00	4.60	5.00
T3: Massivo^®^ (M)	99.0	96.0	5.55	4.00	4.50	6.00
T4: Ídolo^®^ (I)	98.0	97.0	5.51	4.00	4.50	5.00
T5: V + L	99.0	97.0	5.42	4.00	4.60	6.00
T6: V + L + I	98.0	97.0	5.36	4.00	4.70	5.00
T7: V + M	99.0	98.0	5.51	4.00	4.60	5.00
T8: V + M + I	99.0	96.0	5.52	4.00	4.60	6.00
T9: V + I	98.0	97.0	5.62	4.00	4.40	5.00
T10: L + I	99.0	97.0	5.48	4.00	4.60	5.00
T11: I + M	99.0	95.0	5.51	4.00	4.60	6.00
Mean square—Treatment	4.40 ns	116 ns	0.100 ns	-	0.040 ns	0.500 ns
Mean square—Blocks	2.10 ns	203 **	2.10 **	-	1.50 **	2.00 **
Mean square—Residual	3.60	18.50	0.100	-	0.030	0.400
Coefficient of variation (%)	1.90	4.50	4.20	-	3.70	11.7

^1^ Means do not differ from each other when using the Scott–Knott clustering test at a 5% probability. ** and ns significant at 1% probability and not significant when using the F-test, respectively.

**Table 7 plants-12-00338-t007:** Chemical analysis of macro- and micronutrients in soybean and corn seeds.

Description	Macronutrient (g kg^−1^)					Micronutrient (mg kg^−1^)					
	N	P	K	Ca	Mg	S	B	Cu	Fe	Mn	Zn
Soybean	60.3	5.10	20.0	2.20	3.00	2.70	26.9	10.8	78.0	25.1	33.2
Corn	18.9	2.90	3.40	0.200	1.20	1.10	3.90	1.00	59.1	4.80	22.3

N—nitrogen; P—phosphorus; K—potassium; Ca—calcium; Mg—magnesium; S—sulfur; B—boron; Cu—copper; Fe—iron; Mn—manganese; Zn—zinc.

**Table 8 plants-12-00338-t008:** Treatment of soybean NS7667 IPRO and corn MG744 PWU seeds with different mineral fertilizers, formulated products, and respective doses applied according to the weight or number of seeds used per soybean and corn sample.

Dose		
Treatment	Soybean (mL kg^−1^)	Corn (mL 960 seeds^−1^)
Control (intact seed)	-	-
Control (water—W)	5.0 Water (W)	X
Control (washed)	X	-
Control (washed + water)	X	8.0 W
T1: Vital^®^ (V)	1.5 V + 3.5 W	2.4 V + 5.6 W
T2: Lança^®^ (L)	1.0 L + 4.0 W	1.6 L + 6.4 W
T3: Massivo^®^ (M)	1.0 M + 4.0 W	1.6 M + 6.4 W
T4: Ídolo^®^ (I)	1.0 I + 4.0 W	1.6 I + 6.4 W
T5: V + L	1.5 V + 1.0 L + 2.5 W	2.4 V + 1.6 L + 4.0 W
T6: V + L + I	1.5 V + 1.0 L + 1.0 I + 1.5 W	2.4 V + 1.6 L + 1.6 I + 2.4 W
T7: V + M	1.5 V + 1.0 M + 2.5 W	2.4 V + 1.6 M + 4.0 W
T8: V + M + I	1.5 V + 1.0 M + 1.0 I + 1.5 W	2.4 V + 1.6 M + 1.6 I + 2.4 W
T9: V + I	1.5 V + 1.0 I + 2.5 W	2.4 V + 1.6 I + 4.0 W
T10: L + I	1.0 L + 1.0 I + 3.0 W	1.6 L + 1.6 I + 4.8 W
T11: I + M	1.0 I + 1.0 M + 3.0 W	1.6 I + 1.6 M + 4.8 W

For the seed treatment, slurries of 5.0 mL for each kg of soybean seeds and 8.0 mL for every 960 corn seeds were added, constituted of the different mineral fertilizers and water doses.

## Data Availability

Not applicable.

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
