# Peer review of "Germination and Vigor of Soybean and Corn Seeds Treated with Mixed Mineral Fertilizers"

_plants, 2023, doi:10.3390/plants12020338_

Round 1

Reviewer 1 Report

Dear Editor,

Please receive my review for the manuscript entitled “Physiological potential of soybean and corn seeds treated with mixed mineral fertilizers” by  Machado et al. 2022.

The authors studied the effects of seed treatment with mixed fertilizers on some physiological parameters. The authors found that these treatments had minimal effects. The manuscript is well written and information obtained from this experiment is useful for  growers and scientific communities. However, the authors need to address the following points before accepting the manuscript.

-Abstract; The authors need to justify the research (why did they conduct this research; the research need).

-Discussion: The experiment was conducted for one year/one location; therefore, it is not repeated. Therefore, how do we know that these results will be repeated if the environment (location or year) will change? The authors need to include in the discussion that the scope of these results are limited as the experiment was not repeated across years and locations. This point should be included in the conclusion as well. 

Author Response

Point 1: Abstract: The authors need to justify the research (why did they conduct this research; the research need).

Response 1: We agree with the opinion of the supervisor and the text has been improved.

Point 2: Discussion: The experiment was conducted for one year/one location; therefore, it is not repeated. Therefore, how do we know that these results will be repeated if the environment (location or year) will change? The authors need to include in the discussion that the scope of these results are limited as the experiment was not repeated across years and locations. This point should be included in the conclusion as well. 

Response 2: We agree with the opinion of the supervisor and the text has been improved.

Reviewer 2 Report

The manuscript describes effects of seed treatments with mineral fertilisers on the germination and field emergence of two cultivated plants - soybean and corn. A considerable effort was put into testing 11 treatments and collecting data from laboratory germination tests and field experiments for both crops. The obtained results can be interesting to researchers designing seed treatments for corn and soybean even though no spectacular effects were found. The manuscript can be improved in order to make it more understandable to potential readers. Below, I will list a few suggestions.

Abstract: This is the worst part of the manuscript. It is carelessly written and highly confusing. The names of fertilizer products and their combinations in the mentioned treatments are not consistent and written with mistakes. There are also other mistakes like ‘F thesis’ instead of ‘F test’. However, the biggest problem is that this abstract does not summarize well the work that was done, so it should be completely rewritten. I wouldn’t list all 11 treatments because it would be enough to write that 4 commercial fertilisers (here provide their names) were tested separately and in different combinations. Then you could mention selected treatments which were the most effective or detrimental in different experiments. Another problem is that the term ‘physiological potential of seeds’ seems to me a bit ambiguous. In my opinion, it would be better to write more precisely about specific germination parameters (germination percentage etc.) and emergence in the field in the context of the used treatments. For this reason I would also consider changing the title.

Results: There is a lot of data reported here. I would divide this section using additional subheadings.

Results/methods: I don’t understand why table 8 specifies different doses for each treatment and each crop. There is nothing on the effects of the doses of fertilisers in the results, so I expected that only one dose per treatment was used. How much Vital was applied for example on soybean in treatment T1: 1.5 or 3.5 ml per kg of seeds? Please clarify table 8, caption of this table or description of the methods so that readers don’t have to wonder about the used doses.

I think using two controls in soybean experiments and three controls in corn experiments is too complicated. The aim of the experiments was to test effects of the fertilisers so it would be best if control seeds differed from seeds in treatments T1-11 only with application of the fertiliser. If seeds were washed before applying fertiliser then washed seeds without fertiliser should constitute a control and data on intact control seeds should be removed from the tables with results. I leave selection of the best control to the authors.

Cited literature: It would be beneficial to the readers if more literature published in international journals (in English) was cited in the introduction and discussion. About 70% of the references cited in the manuscript doesn’t even have an English title, therefore they may not understandable to a broad audience.

Author Response

Point 1: Abstract: This is the worst part of the manuscript. It is carelessly written and highly confusing. The names of fertilizer products and their combinations in the mentioned treatments are not consistent and written with mistakes. There are also other mistakes like ‘F thesis’ instead of ‘F test’. However, the biggest problem is that this abstract does not summarize well the work that was done, so it should be completely rewritten. I wouldn’t list all 11 treatments because it would be enough to write that 4 commercial fertilisers (here provide their names) were tested separately and in different combinations. Then you could mention selected treatments which were the most effective or detrimental in different experiments. Another problem is that the term ‘physiological potential of seeds’ seems to me a bit ambiguous. In my opinion, it would be better to write more precisely about specific germination parameters (germination percentage etc.) and emergence in the field in the context of the used treatments. For this reason I would also consider changing the title.

Response 1: We agree with the opinion of the supervisor and the text has been improved.

Point 2: Results: There is a lot of data reported here. I would divide this section using additional subheadings.

Response 2: We agree with the opinion of the supervisor and the text has been improved.

Point 3: Results/methods: I don’t understand why table 8 specifies different doses for each treatment and each crop. There is nothing on the effects of the doses of fertilisers in the results, so I expected that only one dose per treatment was used. How much Vital was applied for example on soybean in treatment T1: 1.5 or 3.5 ml per kg of seeds? Please clarify table 8, caption of this table or description of the methods so that readers don’t have to wonder about the used doses.

Response 3: We agree with the opinion of the supervisor and the text has been improved.

Point 4: I think using two controls in soybean experiments and three controls in corn experiments is too complicated. The aim of the experiments was to test effects of the fertilisers so it would be best if control seeds differed from seeds in treatments T1-11 only with application of the fertiliser. If seeds were washed before applying fertiliser then washed seeds without fertiliser should constitute a control and data on intact control seeds should be removed from the tables with results. I leave selection of the best control to the authors.

Response 4: We understand the supervisor's comments and suggetions,  however in the region where the experiment was carried out, the companies only trade corn seeds with crop protection products company standards. So, it is understood that the presence of this extra witness helps on the understanding of the results, having actually an extra condition for the comparison

Point 5: Cited literature: It would be beneficial to the readers if more literature published in international journals (in English) was cited in the introduction and discussion. About 70% of the references cited in the manuscript doesn’t even have an English title, therefore they may not understandable to a broad audience.

Response 5: We agree with the opinion of the supervisor and the text has been improved.
